# Soil Responses to High Olive Mill Wastewater Spreading

Leïla Chaâri [1,*], Norah Salem Alsaiari [2], Abdelfattah Amari [3,4], Faouzi Ben Rebah [5], Monem Kallel [1] and Tahar Mechichi [6,*]

1   Laboratory of Environmental Engineering and Ecotechnology, National School of Engineers of Sfax (ENIS), University of Sfax, P.O. Box 1173, Sfax 3038, Tunisia; monemkallel@gmail.com
2   Department of Chemistry, College of Science, Princess Nourah bint Abdulrahman University, P.O. Box 84428, Riyadh 11671, Saudi Arabia; nsalsaiari@pnu.edu.sa
3   Department of Chemical Engineering, College of Engineering, King Khalid University, Abha 61411, Saudi Arabia; abdelfattah.amari@enig.rnu.tn
4   Research Laboratory of Processes, Energetics, Environment and Electrical Systems, Department of Chemical Engineering & Processes, National School of Engineers, Gabes University, Gabes 6072, Tunisia
5   Higher Institute of Biotechnology of Sfax, University of Sfax, Route de Soukra Km 4, Sfax 3000, Tunisia; benrebahf@yahoo.fr
6   Laboratory of Biochemistry and Enzyme Engineering of Lipases, National School of Engineers of Sfax (ENIS), University of Sfax, P.O. Box 1173, Sfax 3038, Tunisia
*   Correspondence: leila-chaari@hotmail.fr (L.C.); tahar.mechichi@enis.rnu.tn (T.M.)

**Abstract:** Olive manufacturing generates the most polluting wastewater. Olive mill wastewater (OMW) contains a large amount of organic and inorganic fractions. Olive-oil-producing countries have investigated several treatments and valorization processes for better management of this waste. The Tunisian government adopted OMW spreading on soil to manage the waste and improve the organic matter in the soil of olive groves. The examination of soil after OMW spreading was set up to assess the physicochemical changes and better comprehend the soil's responses. An incubation of two types of artificial soil treated with 40 and 80 $m^3 \cdot ha^{-1}$ of OMW led to increased organic matter, phosphorus, nitrogen and potassium contents. The adsorption of the phenolic compounds in soil was dependent on the clay type and was shown by the behavior of the soil composed of bentonite clay. The germination index of tomato and alfalfa seeds recorded a positive test with OMW applied on soil, and it was in relation to the species utilized. This practice seems to be a solution for the management of OMW because it limits the use of chemical fertilizers and might be a convenient source of carbon in organic farming.

**Keywords:** OMW spreading; soil organic matter; phenolic compounds; soil incubation

## 1. Introduction

Tunisia is one of the major world producers of olive oil, with an average production of 217,760 t during the campaign 2014–2018 [1]. The extraction of olive oil requires a high amount of water and, correspondingly, generates a large amount (more than 30 million $m^3$) of waste known as OMW [2,3]. This wastewater is produced during the wintertime, from November to February [4], creating a serious environmental problem. The volume of OMW produced depends on the milling method [5,6]. Three systems are used for olive oil extraction including a press system, a two-phase centrifugal system or a three-phase centrifugal system. The pressure system generates 25 to 35 L of OMW/100 kg of crushed olives while the three-phase technology produces 40 to 45 L of OMW/100 kg of crushed olives [7].

OMW generated by the press system has a low pH and contains high levels of fats and oils (2.8 $g \cdot L^{-1}$), organic matter (OM; near 110.53 $g \cdot L^{-1}$), suspended solids and contaminating compounds such as polyphenols (PC; about 17.15 $g \cdot L^{-1}$) [8]. The techniques applied for the treatment of OMW (physicochemical or biological treatments) are complex

and expensive [9,10]. In the last two decades, biotechnological techniques have been proposed and tried for the OMW treatment [11,12]. Though this technique has encouraging results, it is not utilized in the common practice. Therefore, there are other options for managing the OMW, for example, biogas [13,14], bioactive molecule production [15,16] and composting [17].

A feasible option is OMW land spreading, which is based on controlled doses of this effluent to improve the soil. This option could be used as a cheap soil conditioner and/or fertilizer [18] to solve the problem of the chronic water scarcity affecting Mediterranean agricultural areas [19]. This practice has an environmental and economic benefit but should be applied with caution. S'habou et al. [20] and Paredes et al. [21] already showed that the agronomic reuse of OMW without following appropriate protocols for soil application could degrade its characteristics. The spreading of OMW on soil has been largely investigated [22,23]. El Hassani et al. [24] tackled the contribution of OMW and soil microbial groups to OMW organic matter humification in soil. Chartzoulakis [25] showed that, following 3 years of raw OMW application, there were no significant differences in pH, electrical conductivity (EC), phosphorus (P), sodium and organic levels between the control and OMW-treated soils. Furthermore, researchers showed an improvement in olive yield after the progressive use of OMW on soil for 6 years [26]. A further benefit of OMW application is the increase in the soil's aggregate stability [27].

Therefore, extensive investigations focused on the change in salinity, pH and hydraulic conductivity and on the accumulation of phytotoxic polyphenolic compounds inhibiting the soil's microbial activity [28,29]. Another study showed the inhibition of the arbuscular mycorrhizal fungal root colonization by phenolic compound fraction in reducing the nutrient uptake of the olive trees [30]. Di Bene et al. [22] showed that long-term repeated OMW spreading has no remaining impacts or negative trends on the soil's chemical and biochemical changes. Nevertheless, this practice has recently created a controversy over the fertilization properties of the soil and the impacts related to its acidity, salinity, organic matter and phenolic compounds. Under Mediterranean conditions, the OMW contains a residual oil that could become hydrophobic once irrigated on soils [31,32]. Regarding the hydrological characteristics of soils applied with OMW, much alert could be paid to the conceivable decrease in water infiltration. In Mediterranean areas, where the infiltration-excess mechanism dominates the soil's hydrological response [33], a reduced infiltration capacity could make these areas particularly prone to runoff and soil erosion risks [34]. Bombino et al. [35] demonstrated that land spreading with OMW does not significantly change soil water repellency. This is why the preservation and enhancement of soil organic characteristics should be one of the priorities in the near future in order to restore soil fertility and yields of marginal and degraded croplands [36]. From one viewpoint, water scarcity and low soil fertility in olive-producing countries lead to reutilizing OMW for irrigation and fertilization of the soils in Tunisia and other Mediterranean countries. The light texture of sandy soils affects the soil water's downward movement to its amplified losses. Accordingly, the OMW application plays an efficient role in water maintenance and limitation of losses to deeper sandy soil layers. An increase in the organic matter of OMW sites showed that a regular application of OMW for 5 and 15 years increased the soil's aggregate stability [27].

This study evaluated the short-term effects (3 months) of the OMW application on artificial soils, which were sandy clay loam soils made at a laboratory scale. We hypothesized that the high contents of OM, nitrogen and potassium levels in OMW may noticeably improve the soil's fertility.

## 2. Materials and Methods

Raw OMW was collected from an evaporation pond located in Agareb (Sfax, Tunisia) during summertime, stored at 4 °C and then characterized accordingly before application. The OMW characterization was determined by standard methods in triplicate. The electrical conductivity (EC) and pH were measured using conductivity meter and a pH

meter, respectively. The chemical oxygen demand (COD) was determined according to AFNOR T 90-101. Total phosphorus (P) was measured colorimetrically [37], and total nitrogen (N) was determined by the Kjeldahl method (AFNOR T 90-101). The OM was measured after the incineration of samples at 550 °C for 4 h and $K^+$, $Na^+$, $Ca^{2+}$ and $Mg^{2+}$ by atomic absorption spectrophotometry (ThermoFisher Scientific ice 3000 Series, Waltham, Massachusetts, USA). The total phenolic compound (PC) value was determined using the Folin–Ciocalteu method [38]. This result was expressed in ppm by reference to a standard curve using a pyrogallol solution. Chlorides ($Cl^−$) and sulfates ($SO_4^{2−}$) were determined by ion chromatography (Schimadzu model HIC-6A, Colombia, MD, USA).

The artificial soil samples were put in bins (20 × 20 cm) and were prepared as described in OECD [39]. These soils were composed of 70% sand, 20% clay and 10% peat. For our study, we formed two artificial soils by changing the clay type (soil 1 and soil 2 were prepared with kaolinite and bentonite, respectively). After spreading the doses of 40 and 80 $m^3 \cdot ha^{−1}$ of OMW on soil, the bins were placed in an incubator at 20 °C for 12 weeks, and soil samples were taken every 2 weeks. The first day of incubation was taken as day = 0 for the experimentation. The bins designated D40 and D80 were respectively irrigated with doses of 40 and 80 $m^3 \cdot ha^{−1}$ of raw OMW. The bin that was not irrigated with OMW (D0) served as a control soil. The soil samples were stored at −20 °C prior to analysis. For soil analysis, pH and EC were measured on a mixture of soil–water 1:5. The SOM was determined after incinerating the samples at 550 °C for 4 h, P and total nitrogen as mentioned above. The extraction of exchangeable bases (Na, K, Ca and Mg) was determined according to the procedure described by Pauwels et al. [40]. A mixture of soil–water 1:5 followed by ion chromatography was utilized to determine $Cl^−$ and $SO_4^{2−}$ levels [41]. Phenolic compounds were extracted with sodium pyrophosphate 0.4 N and sodium hydroxide 0.1 N [7] and quantified by the Folin–Ciocalteu method as mentioned above.

In this investigation, UV-visible spectroscopy was adopted following the strategy proposed by Zbytniewski and Buszewski [42]. A combination of air-dried soil (1 g) and a volume (50 mL) of NaOH (0.5 M) was shaken for 2 h and afterward centrifuged at 3000 rpm for 25 min. The absorbance of the supernatant was recorded on a spectrophotometer examining it in the range of 200 to 1000 nm. To recognize boundaries appropriate for describing the absorption spectra, E6, E4 and E2 were determined, where E was the optical density of the solution, and the attached lists 6, 4 and 2 related to 665 nm, 465 nm and 280 nm [43]. The ratios E2/E4, E2/E6 and E4/E6 were determined to express the intensity of the humification process. The ratio E2/E4 was utilized as an indicator of the beginning of humification. The ratio E2/E6 reflected the relation between non-humified and highly humified materials. At last, the E4/E6 ratio recorded the highest humification degree.

A phytotoxicity test was conducted on soil sample extracts. The germination index (GI) of tomato (*Lycopersicum esculentum* L.) seeds and that of alfalfa (*Medicago sativa*) seeds were determined according to the method proposed by Mari et al. [44]. The GI was determined as follows: a soil extract was prepared with the ratio of 1/10 (soil/deionized water), stirred for 2 h and centrifuged at 9000 rpm. Tomato and alfalfa seeds were distributed on a filter paper in Petri dishes and moistened with 5 mL of a soil extract. The soil samples were incubated at 25 °C for 5 days. The GI was calculated using the following formula:

$$GI = 100 \times (Gs/Gc) \times (Ls/Lc)$$

where Gs and Gc were germinated seeds in the sample and control soils, and Ls and Lc were the mean root elongation in the sample and control, respectively. GIT and GIA were attributed to GI of tomato and alfalfa, respectively.

For the statistical analysis, soil chemical parameters and a phytotoxicity test were presented as average values of three replicates. A basic statistical analysis of the data was performed using Microsoft Excel program for Windows. To build the relationship between chemical soil parameters and GI after OMW spreading on soil, a principal component analysis (PCA) was adopted. PCA, a correlation method based on the principal component

scores, changed the data of many tested variables into a set of compound axes. The determined correlation matrix (Pearson correlation coefficient value $p < 0.05$) showed that the parameters with a significant correlation could be classified into a group. The software used for the PCA study was XLSTAT 2014, a complement to the Excel software 2013.

## 3. Results

### 3.1. OMW Characteristics

OMW chemical proprieties rely on the ripeness of olive variety, climate, soil conditions and the oil extraction technique. Selected parameters of OMW applied on artificial soils are given in Table 1. OMW was acidic at pH 5.33 with a high organic load (OM = 210.55 g·L$^{-1}$ and COD = 154.9 g O$_2$·L$^{-1}$) and a level of PC of about 4.65 g·L$^{-1}$. Moreover, OMW was characterized by a high level of K and Na with an average of 7716 ppm and 3490 ppm, respectively.

**Table 1.** OMW characterization applied to soils.

| Characteristics | Mean Value |
|---|---|
| pH | 5.33 |
| EC (mS·cm$^{-1}$) | 18.58 |
| COD (g O$_2$·L$^{-1}$) | 154.9 |
| OM (g·L$^{-1}$) | 210.55 |
| N (g N·L$^{-1}$) | 0.432 |
| P (g P·L$^{-1}$) | 0.65 |
| PC (g·L$^{-1}$) | 4.62 |
| Na$^+$ (ppm) | 3490 |
| K$^+$ (ppm) | 7716 |
| Ca$^{2+}$ (ppm) | 1028.5 |
| Mg$^{2+}$ (ppm) | 541 |
| Cl$^-$ (ppm) | 2058 |
| SO$_4^{2-}$ (ppm) | 840 |

### 3.2. Mineral Fraction Evolution after OMW Spreading on Soil

As a vital component of soil fertility, the soil's chemical property reflects its potential ability to provide nutrients for plants [45]. The interest in this aspect surpassed the increase in soil fertility since it was involved in environmental issues such as salinization and solidification of soils. The soil and peat characteristics are summarized in Table 2.

**Table 2.** Soil and peat characteristics.

| | pH | CE (µS·cm$^{-1}$) | CaCO$_3$ (%) |
|---|---|---|---|
| Soil 1 | 7.12 | 570 | 4.25 |
| Soil 2 | 8 | 900 | 10.75 |
| Peat | 7.2 | 420 | - |

### 3.2.1. Effect of OMW on pH Soil and EC Progress

The pH of soils 1 and 2 after OMW spreading with doses of 40 and 80 m$^3$·ha$^{-1}$ (Figure 1) showed an increase from 7 to 9 after an incubation period (IP). For soil 1, pH varied between 7 and 7.15 for both doses. At the beginning of the incubation period, the pH of control soil was 7.12, 7.15 for soils that received the dose of 40 m$^3$·ha$^{-1}$ and 7.03 for soils that received 80 m$^3$·ha$^{-1}$. Additionally, the curve D80 at IP = 2 weeks reached 8.22 and then tended to decrease gradually until reaching 7.7 after two months. From the 8th week, the soil pH stabilized showing the buffering impact of soils. For soil 2, the initial pH was alkaline in the order of 8. Following OMW spreading at the doses of 40 and 80 m$^3$·ha$^{-1}$, the soil's pH varied slightly but still remained alkaline close to 8. In fact, soil 2 was made of bentonite clay and presented a high percentage of CaCO$_3$ of around 10.79% (Table 2).

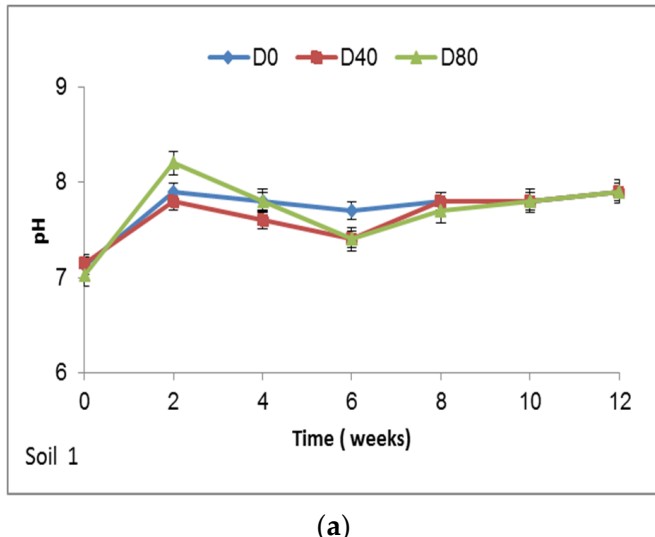 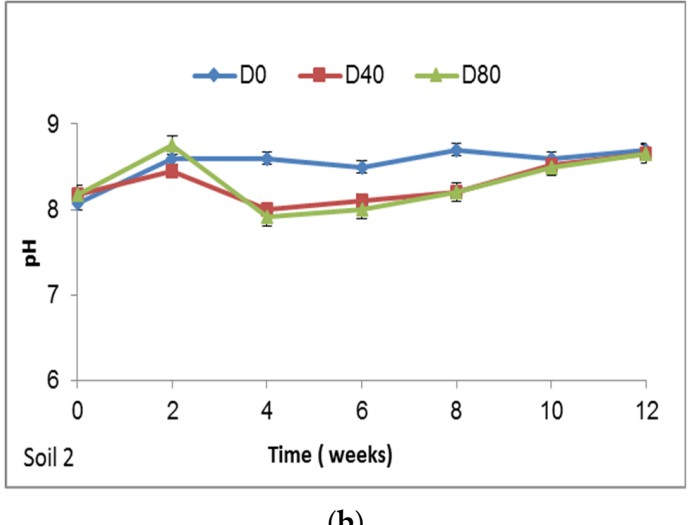

(**a**)  (**b**)

**Figure 1.** Effect of OMW on pH soil 1 (**a**) and soil 2 (**b**).

During the IP, EC values of soils 1 and 2 receiving OMW varied between 550 and 1460 $\mu S \cdot cm^{-1}$ (Figure 2). These EC variations increased depending on the dose applied on soils. Upon OMW application on soil 1, EC values varied from 570 to 705 to 860 $\mu S \cdot cm^{-1}$ following the applied doses of 0, 40 and 80 $m^3 \cdot ha^{-1}$, respectively. The D0 and D40 curves had similar distributions from the 6th week. The D80 curve recorded the highest EC changes throughout the IP. Figure 2 related to soil 2 shows that from the 4th week of incubation, the changes in the EC of soil were comparable for the doses of 0 and 40 $m^3 \cdot ha^{-1}$. The EC variations for the soil treated with a dose of 80 $m^3 \cdot ha^{-1}$ showed values decrease from 1455 $\mu S \cdot cm^{-1}$ at IP = 0 weeks to 1184 $\mu S \cdot cm^{-1}$ at IP = 12 weeks. These results confirm that soil EC decreased over time. Indeed, soil EC variations showed high values during the incubation for soil 2 due to the presence of bentonite, and the evolution of this parameter over time (12 weeks) remained at a relatively high level. On the other hand, for soil 1 containing kaolinite, EC variations tended to decrease gradually over time and stabilize at values close to control.

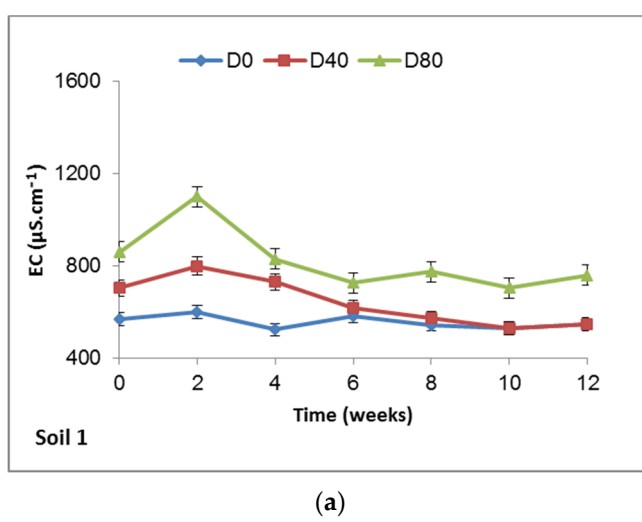 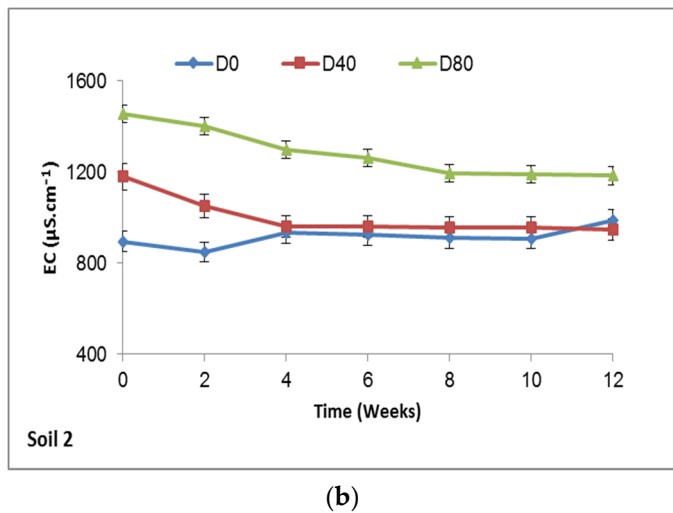

(**a**)  (**b**)

**Figure 2.** Soil 1 (**a**) and soil 2 (**b**) EC progress after OMW amendment.

### 3.2.2. N, P and K Evolution

The study of N dynamics in soil (Figure 3a,b) showed that N levels decreased compared to the control soil from the second week of incubation. From the fourth week of

incubation, N levels increased non-proportionally to the applied doses of OMW, and these recorded rates were higher than rates in the control soil. This evolution was practically comparable for the two soils.

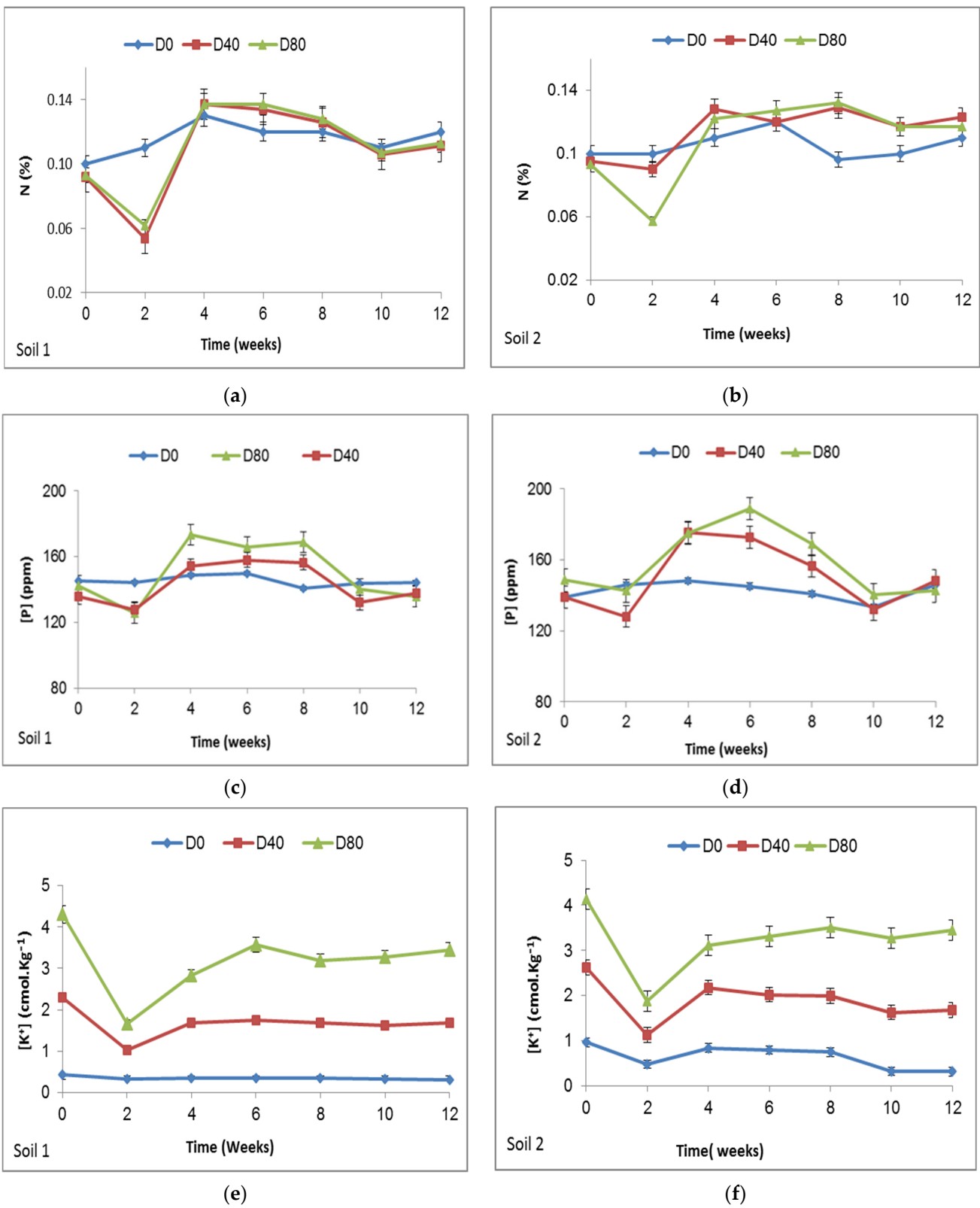

**Figure 3.** N (**a,b**), P (**c,d**) and K (**e,f**) level evolution after OMW application on soils.

The phosphorus level in OMW was about 0.65 g·L$^{-1}$. Figure 3c,d show P evolution in soils after spreading OMW. The result reveals a progress of P similar to N. At IP = 2 weeks, P levels of treated soil 1 and 2 decreased compared to the control soil. From the 4th to 8th weeks, P recorded high concentrations for the soils treated with doses of OMW compared to control. The highest P levels were recorded for the two soils applied with 80 m$^3$·ha$^{-1}$ of OMW.

K recorded a concentration in OMW of about 7716 ppm. Figure 3e,f display K changes after the OMW application on soils 1 and 2. These figures show a notable improvement in K levels in treated soils. The K enrichment was linked to its high level in OMW and was due to the small size of K in hydrated form compared to other ions such as Ca, Mg and Na [46]. The sharp increase in K level was noted throughout the IP proportionally to the doses applied. Soil 1 at IP = 0 weeks receiving the dose of 40 and 80 m$^3$·ha$^{-1}$ of OMW recorded a K concentration 5 and 10 times higher than that of control soil, respectively. The same progress was noticed for soil 2 (IP = 0 weeks). K levels were registered 3 and 4.5 times higher than those of the control soil. The difference in K rates between soil responses was associated with the clay type.

### 3.2.3. Na, Ca and Mg Variation

The Na evolution (Figure 4a,b) in the soil treated with OMW showed that the highest Na concentrations were recorded in the different types of soils. A sudden increase in Na levels in the treated soil 1 was noticed, and these amounts were multiplied by 2 and 2.5, respectively, after the application of OMW doses of 40 and 80 m$^3$·ha$^{-1}$ compared to the control soil. However, the accruement of these levels was not remarkable for treated soil 2, and the difference in the contents was 1.29 and 1.36 cmol·kg$^{-1}$, respectively, for the gradual OMW doses applied. In the second week, a decrease in Na content was registered in treated soils, and it evolved similarly in the control soil. The highest sodium level in soil was recorded at PI = 0 weeks at the level of curve D80 for soil 1. The highest concentration of Ca in soil was noticed for the control soil. The Ca evolution (Figure 4c,d) decreased after 2 weeks of OMW application and then increased at the end of incubation. For example, for soil 1 (IP = 2 weeks), Ca concentration in the control soil was about 12.38 cmol·kg$^{-1}$ and decreased to 9.22 and 9.25 cmol·kg$^{-1}$ respectively to the applied doses of 40 and 80 m$^3$·ha$^{-1}$ of OMW. The Ca levels of treated soils recorded similar values compared to the control soil from the 8th week of incubation. The application of OMW on soils affected the Mg levels (Figure 4e,f). The Mg level of the treated soil 1 decreased compared to the concentrations recorded in the control soil, and it was valid throughout the IP. The treated soil 2 registered an increase in the Mg contents compared to the control soil. During IP = 8 weeks, the soil Mg concentration increased from 7.2 to 8.5 to 8.8 cmol·kg$^{-1}$, respectively, for the doses of 0, 40 and 80 m$^3$·ha$^{-1}$. This increment was not proportional to the OMW doses applied.

### 3.2.4. Cl$^-$ and SO$_4{}^{2-}$ Changes

The results noted in Figure 5a,b show that the chloride soil contents increased gradually throughout the spreading process compared to the control soils. This increase was related to the applied doses. At IP = 0 weeks, the Cl$^-$ levels in soil 1 started with a value equal to 185 ppm in control soil, reaching values of 270 and 325 ppm, respectively, for the doses of 40 and 80 m$^3$·ha$^{-1}$. From the 8th week of incubation, the soil Cl$^-$ levels decreased gradually until reaching, in the case of the treated soil 2, levels close to the control soil. The highest Cl$^-$ concentration was recorded for the dose of 80 m$^3$·ha$^{-1}$, and it was found for the two soils. OMW contained a sulfate concentration of about 840 ppm (Table 1); subsequently, the study of the dynamics of this ion in soils treated with doses of 40 and 80 m$^3$·ha$^{-1}$ was useful. The results of soil SO$_4{}^{2-}$ levels are displayed in Figure 5c,d. At the beginning of the treatment, the responses showed that both treated and untreated soils registered similar concentrations. From the 2nd week of incubation, the sulfate levels of the treated soils decreased considerably compared to the levels of the control soil.

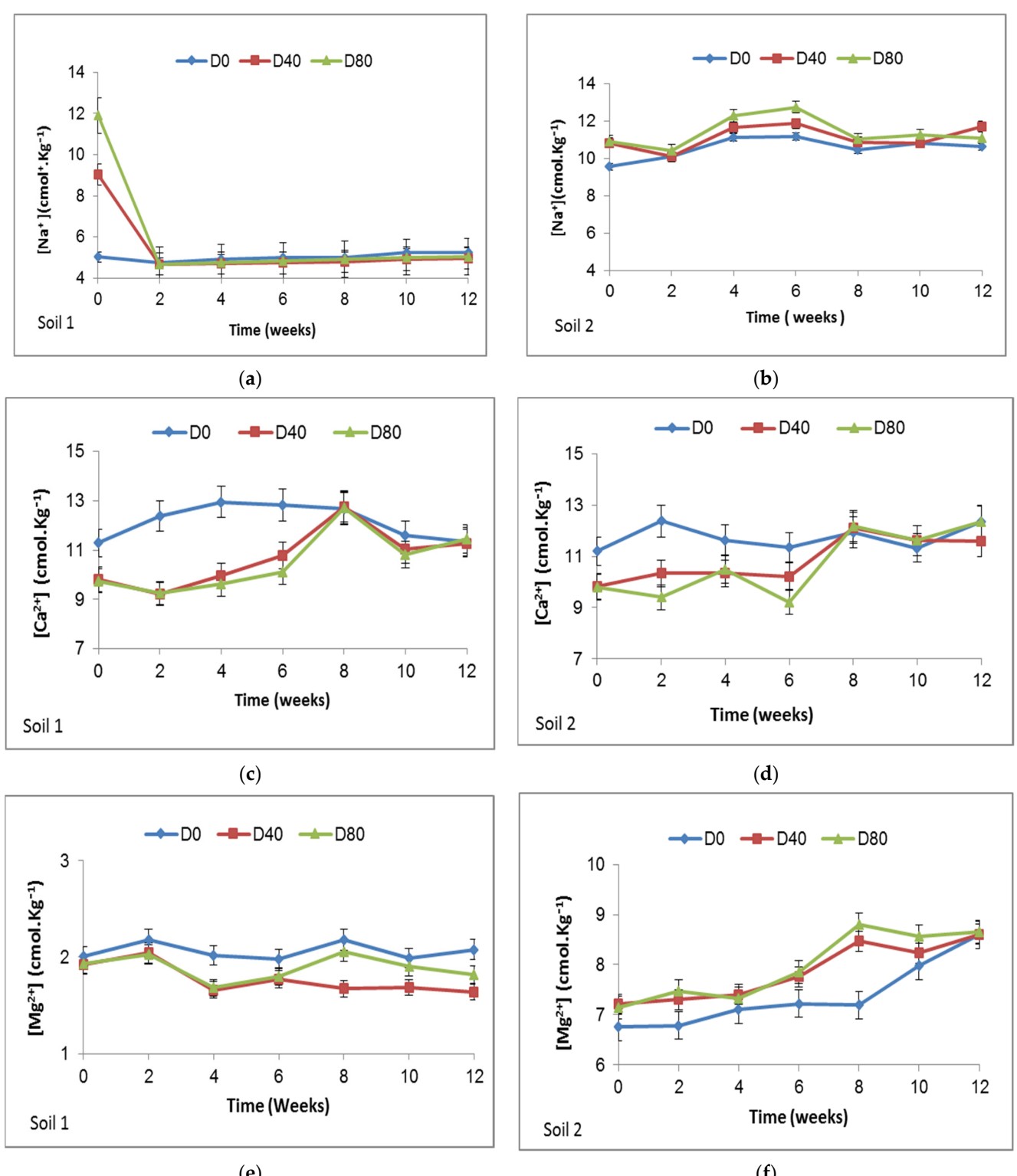

**Figure 4.** Evolution of Na (**a**,**b**), Ca (**c**,**d**) and Mg (**e**,**f**) in soils treated with 40 and 80 m$^3$·ha$^{-1}$ of OMW.

*3.3. Organic Fraction Dynamic after OMW Spreading on Soil*

3.3.1. Organic Matter Evolution

At the beginning of the experiment (IP = 0 weeks), the SOM rates of treated soils (Figure 6) were higher than the values recorded in control soil. Throughout the incubation, the SOM rates of treated soils decreased for a few weeks, and then values exceeded the SOM rates of control soil. Treated soil 1 required 8 weeks for SOM levels to increase compared to

control soil. At the end of the incubation of soil 2, the SOM rates increased from 8.5% to 8.64 to 8.73% for soil applied with OMW doses of 0, 40 and 80 $m^3 \cdot ha^{-1}$, respectively.

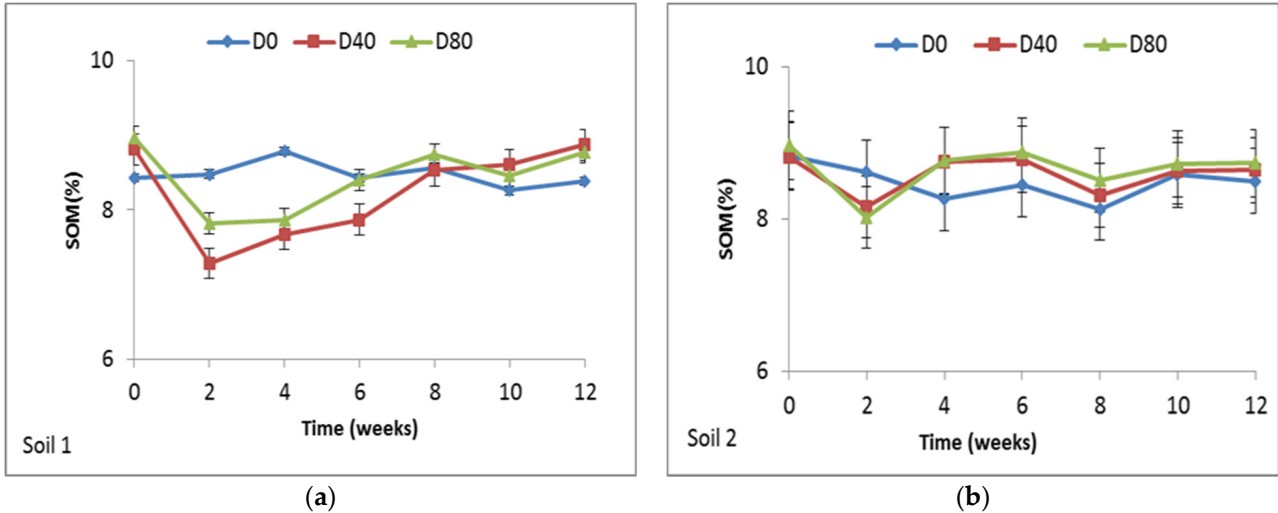

**Figure 5.** Evolution of $Cl^-$ (**a**,**b**) and $SO_4^{2-}$ (**c**,**d**) contents after application of the doses: 0, 40, 80 $m^3 \cdot ha^{-1}$ on soils 1 and 2.

**Figure 6.** Organic matter evolution of amended soil 1 (**a**) and soil 2 (**b**) with OMW.

### 3.3.2. UV Absorption of Humic Substances

The ratios E2/E4, E2/E6 and E4/E6 were determined for soils 1 and 2 (Figure 7). The E2/E4 ratios of treated soils recorded higher values compared to the control soil. At the beginning of the incubation (IP = 0 weeks), the E2/E4 ratio of treated and untreated soils was low under the depolymerization effect. The ratio E2/E6 decreased for both the control and the treated soils. The lowest E2/E6 ratio was recorded in the control soil, and the highest value was noticed in the treated soils with 80 m$^3$·ha$^{-1}$ of OMW. This ratio increment was related to the applied dose. At IP = 0 weeks, a high E4/E6 ratio was registered in treated soils. During the incubation, the E4/E6 ratio decreased considerably.

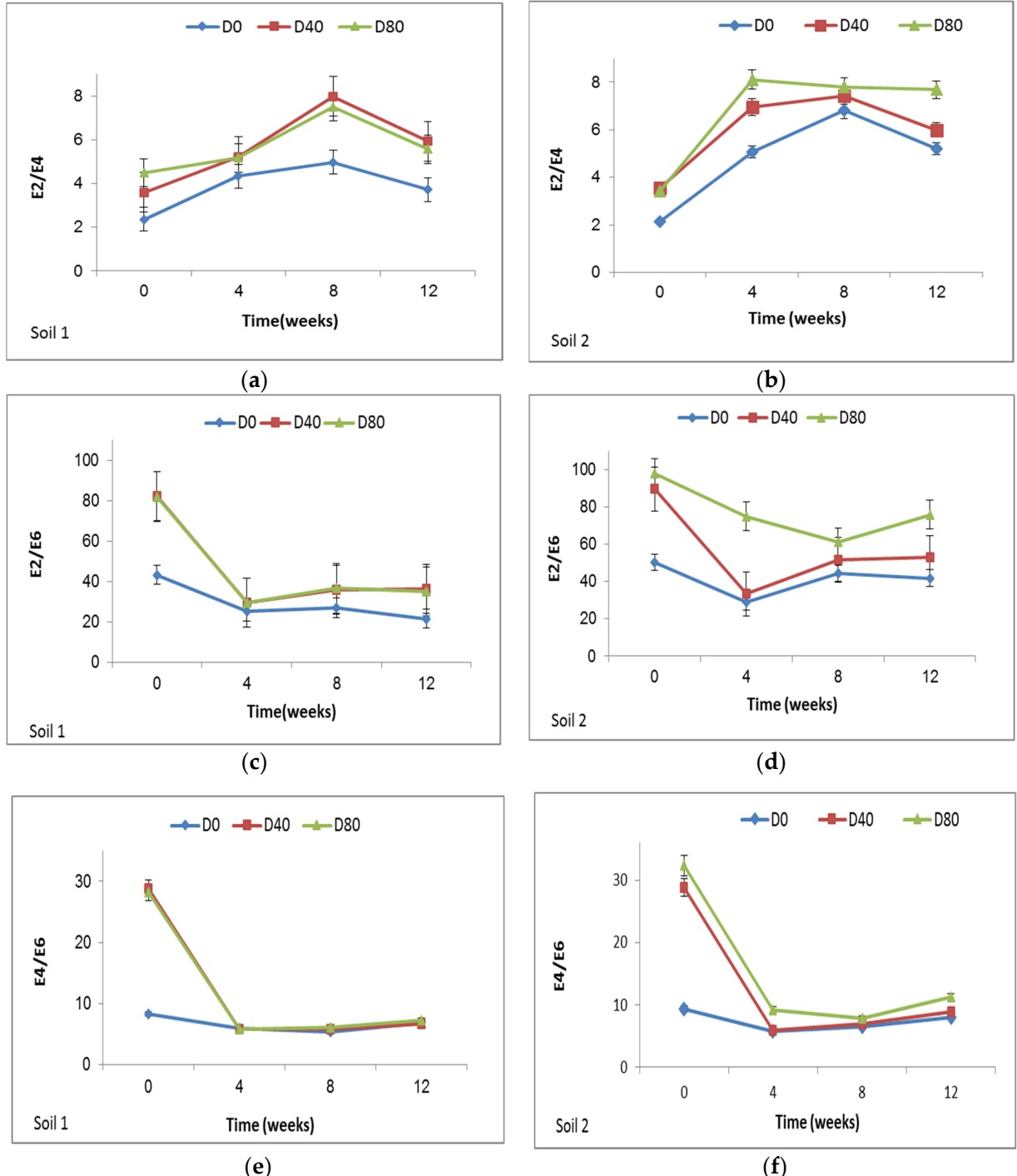

**Figure 7.** Absorption of humic substances in soil 1 (**a**,**c**,**e**) and soil 2 (**b**,**d**,**f**) treated with OMW.

### 3.3.3. PC Evolution

The extraction of PC from soils for quantification was important. The results are illustrated for the treated and untreated soils in Figure 8. For soil 1, the application of OMW amplified the PC concentrations according to the dose applied. At PI = 0 weeks, the PC level increased from 4091 ppm to 5076 to 5879 ppm, respectively, with the gradually applied doses of 0, 40 and 80 $m^3 \cdot ha^{-1}$. The lowest PC level was recorded in the 2nd week. Then, the variation levels of the treated soils were similar to that shown for the control soil. At the start of spreading soil 2 with OMW (PI = 0 weeks), the PC contents of treated soils increased with the applied dose. In the 2nd week of incubation, a low PC level was registered too. Hence, soils 1 and 2 were able to adsorb PC with a higher adsorption capacity for soil 2.

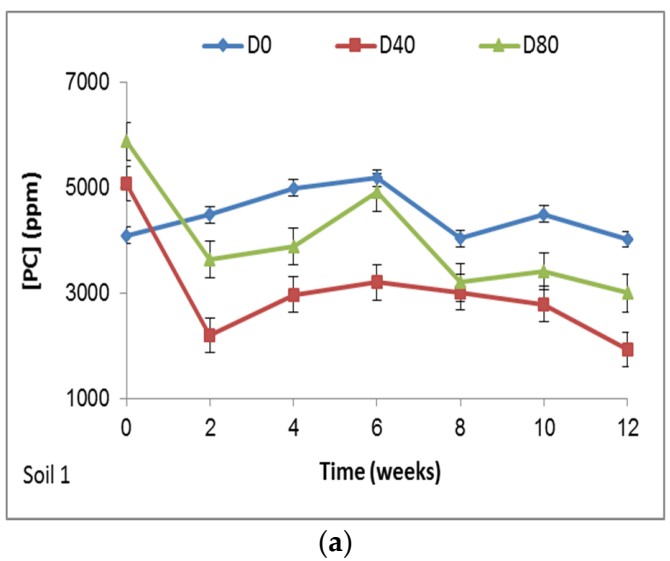

(**a**)

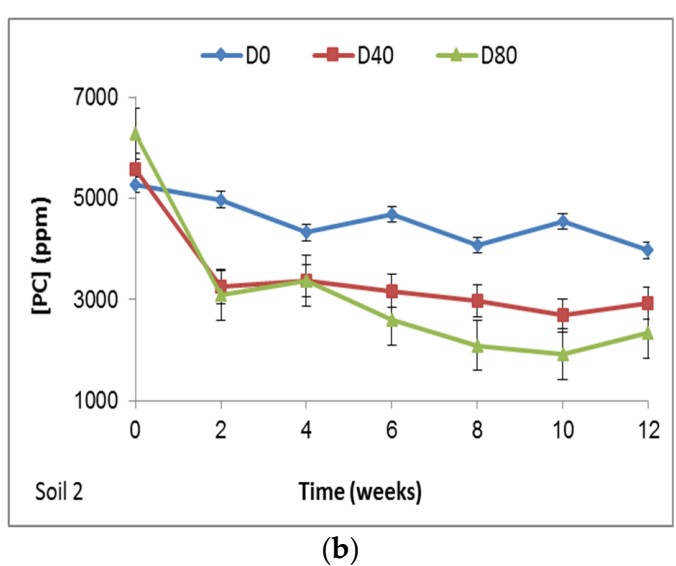

(**b**)

**Figure 8.** Evolution of PC concentrations in soil 1 (**a**,**b**).

### 3.4. Phytotoxicity Test

The percentage of the germination index of alfalfa seeds (GIA) was greater than 150% for both the control and treated soils (Figure 9a,b). At IP= 0 weeks, GIA of the treated soils was lower than that of the control soil. For example, for soil 2, GIA reached 242, 155 and 169%, respectively, with OMW doses of 0, 40 and 80 $m^3 \cdot ha^{-1}$ applied on soil. For soils 1 and 2, the GIA of the treated soil overcame the values registered for the control soil from the 2nd week of incubation.

The result of the percentage of the germination index of tomato seeds (GIT) is illustrated in Figure 9c,d. GIT for treated soil 1 reached 96% and 91% for the respective doses of 40 and 80 $m^3 \cdot ha^{-1}$. Further, from the 2nd week until the 12th week of incubation, the GIT of treated soils exceeded that of the control soil.

### 3.5. PCA Statistical Analysis

Different dynamic parameters (pH, EC, SOM, P, NTK, Na, K, Mg, Ca, PC, GIA and GIT) were studied statistically using a principal component analysis (PCA). The PCA results for soil 1 (Figure 10a) show component 1 (F1) and component 2 (F2) of about 31.56% and 24.59%, respectively. This analysis presented two distinct groups of variables. The first group was formed by the K and EC contents, which were opposite to NTK and Ca levels. The second group was formed by Na and PC concentrations that were opposite to variations of pH, GIA and GIT.

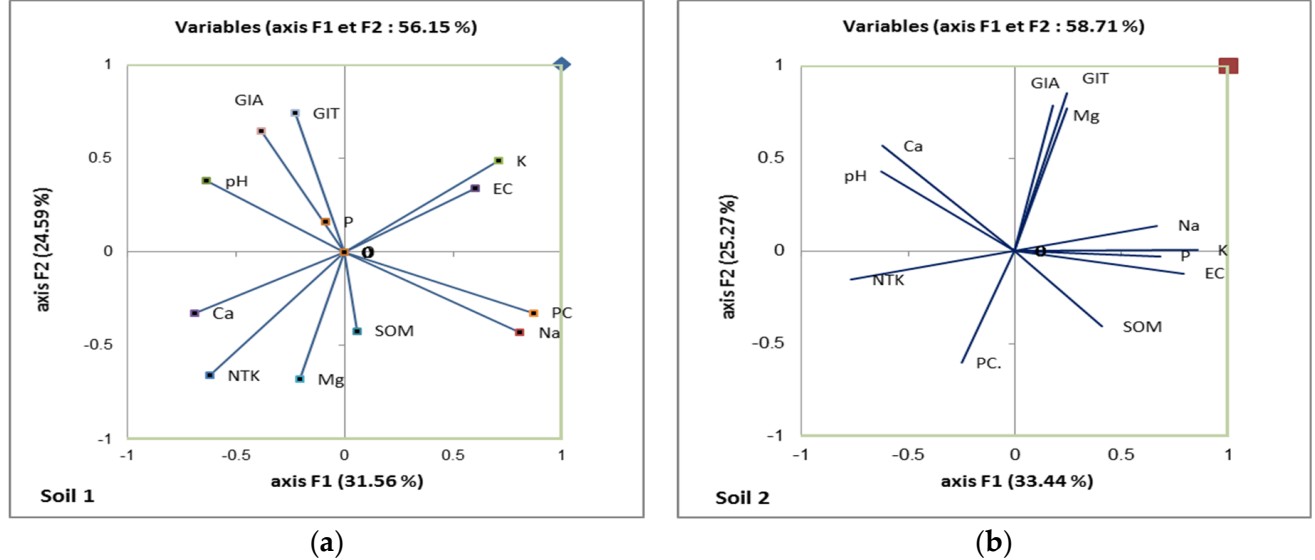

**Figure 9.** Evolution of GIA (**a**,**b**) and GIT (**c**,**d**) after soil treatment with doses of 40 and 80 m$^3 \cdot$ha$^{-1}$.

**Figure 10.** PCA of chemical properties of soil 1 (**a**) and soil 2 (**b**).



For soil 2, PCA results (Figure 10b) display component 1 (F1) at 33.44% and component 2 (F2) at 25.27%. This analysis showed three distinct groups of variables. The first group included the Na, P and K concentrations and the EC contents which were opposite to the NTK. The second group was formed by pH and Ca, which were opposite to the SOM rate. The third group was formed by the Mg content and variations of GIA and GIT opposite to PC levels.

According to PCA, correlations were determined between chemical parameters of soil responses after spreading OMW. Table 3 indicated that pH was significantly positive and was correlated with GIA (r = 0.640) while this parameter was negatively correlated with Na (r = −0.655) and with PC (r = −0.515). EC was significantly negative and was correlated with NTK (r = −0.490) and Ca (r = −0.677) while it was significantly positive and correlated with K (r = 0.570). NTK was positive and significantly correlated with Ca (r = 0.548) and Mg (r = 0.604). PC was negatively correlated with GIA (r = −0.442). Na was correlated positively with K (r = 0.435) and PC (r = 0.877).

**Table 3.** Correlation matrix between chemical parameters of soil 1 amended by OMW.

| Parameters | pH | CE | SOM | P | NTK | Na | K | Ca | Mg | PC | GIT | GIA |
|---|---|---|---|---|---|---|---|---|---|---|---|---|
| pH | 1 | | | | | | | | | | | |
| CE | 0.050 | 1 | | | | | | | | | | |
| SOM | −0.018 | −0.109 | 1 | | | | | | | | | |
| P | −0.216 | 0.079 | −0.447 * | 1 | | | | | | | | |
| NTK | 0.177 | −0.490 * | 0.374 | 0.023 | 1 | | | | | | | |
| Na | 0.655 * | 0.228 | 0.372 | −0171 | −0.208 | 1 | | | | | | |
| K | −0.306 | 0.570 * | −0.045 | 0.242 | −0.690 * | 0.435 | 1 | | | | | |
| Ca | 0.225 | −0.677 * | 0.064 | 0.213 | 0.548 * | −0.306 | −0.433 | 1 | | | | |
| Mg | 0.084 | 0.000 | 0.277 | −0.261 | 0.604 * | 0.056 | −0.420 | 0.287 | 1 | | | |
| PC | −0.515 * | 0.429 | 0.283 | −0.368 | −0.345 | 0.877 * | 0.447 * | −0.472 * | 0.108 | 1 | | |
| GIT | 0.395 | −0.057 | 0.30 | −0.029 | −0.312 | −0.426 | 0.347 | −0.017 | −0.400 | −0.362 | 1 | |
| GIA | 0.640 * | 0.102 | −0.007 | −0.230 | −0.120 | −0.444 | 0.014 | −0.009 | −0.227 | −0.442 | 0.589 * | 1 |

* $p < 0.050$.

For soil 2, Table 4 shows that pH was significantly positive and was correlated with Ca (r = 0.485). On the other hand, it was negatively correlated with K (r = −0.510) and P (r = −0.680). The P content was significantly correlated with K (r = 0.450) and Na (r = 0.773). The PC concentration was significantly negative and was correlated with GIT (r = −0.522) and GIA (r = −0.716). Na was significantly correlated with K (r = 0.499).

**Table 4.** Correlation matrix between chemical parameters of soil 2 amended by OMW.

| Parameters | pH | CE | SOM | P | NTK | Na | K | Ca | Mg | PC | GIT | GIA |
|---|---|---|---|---|---|---|---|---|---|---|---|---|
| pH | 1 | | | | | | | | | | | |
| CE | −0.250 | 1 | | | | | | | | | | |
| SOM | −0.309 | 0.412 | 1 | | | | | | | | | |
| P | −0.680 * | 0.245 | 0.039 | 1 | | | | | | | | |
| NTK | 0.279 | −0.802 * | −0.229 | −0.274 | 1 | | | | | | | |
| Na | −0.428 | 0.293 | −0.037 | 0.773 * | −0.216 | 1 | | | | | | |
| K | −0.510 * | 0.798 * | 0.183 | 0.458 * | −0.750 * | 0.499 * | 1 | | | | | |
| Ca | 0.485 * | −0.572 * | −0.701 * | −0.318 | 0.327 | −0.332 | −0.364 | 1 | | | | |
| Mg | 0.201 | 0.087 | −0.370 | 0.090 | −0.190 | 0.302 | 0.291 | 0.339 | 1 | | | |
| PC | 0.020 | 0.105 | −0.085 | −0.313 | 0.110 | −0.257 | 0.075 | 0.033 | −0.273 | 1 | | |
| GIT | 0.223 | 0.145 | 0.000 | −0.034 | −0.490 * | 0.093 | 0.130 | 0.236 | 0.497 * | −0.522 * | 1 | |
| GIA | 0.246 | 0.090 | −0.037 | 0.092 | −0.225 | 0.229 | 0.174 | 0.43 | 0.619 * | −0.716 * | 0.656 * | 1 |

* $p < 0.050$.

## 4. Discussion

For pH progress, the slight decrease in the pH of soil 1 could be explained by the presence of some organic acids brought by the OMW (pH 5.33). Soil 2 pH varied slightly but was still close to 8 due to the buffering effect and the composition of soil. The pH evolution of the soils revealed that the acidity of OMW could not be a major detrimental factor for

applying OMW in soil at acceptable doses. In addition, Di Bene et al. [22] mentioned that the pH values of the soil having received 80 $m^3 \cdot ha^{-1}$ reached values similar to those of the control soil (0 $m^3 \cdot ha^{-1}$), after six months of the spreading process. In the short term, Piotrowska et al. [47] showed that soil pH decreased from 8.3 to 7.4 after applying a dose of 80 $m^3 \cdot ha^{-1}$ of OMW. The decrease in the pH may be due to the effect of a biochemical reaction based on the ammonium conversion, and subsequently, ammonium oxide produced nitric acid or nitrate [48,49]. Barbera et al. [19] revealed that the decrease in soil pH was due to acids delivered from lipid hydrolysis when OMW was applied on soils with low cation exchange capacity (CEC). In a long-term application, Mechri et al. [2] concluded that the OMW acidity was neutralized by carbonate on the top soil layer. Other research groups found similar results by applying OMW on clay, limestone and carbonate soils [50–53]. After a few weeks of spreading, Piotrowska et al. [47] showed that OMW had a temporary influence on the soil pH, and this fluctuation was not significant after applying the dose of 420 $m^3 \cdot ha^{-1}$ even during an amendment of three successive years [25]. The pH variations could not be the cause of an imbalance in the soil biotope.

The EC gave a clear idea about OM mineralization and mineral element immobilization. The EC increment of the treated soils was linked to the high salt load of OMW (18.58 $mS \cdot cm^{-1}$). The same EC evolutions were observed in the studies of López-Piñeiro et al. [54,55]. Sierra et al. [6] established that soil EC values were proportional to the rates of the applied effluent. For a period of more than two months, the EC of soil treated with an 80 $m^3 \cdot ha^{-1}$ dose increased at the layer 0–20 cm [53]. After 2 weeks of incubation, the EC of soil stabilized as reported by Piotrowska et al. [29]. The stability of soil EC variations changed from one soil type to another; this might be related to the clay mineralogy especially when Feller and Beare [56] mentioned that the formation of a stable soil is directly influenced by the mineralogy, texture, quantity and quality of amended OM.

In fact, the N loss during the second week of incubation may be due to the nitrogen volatilization into $NH_3$ and/or the nitrification process [57,58]. Other studies showed that the presence of polyphenols affected nitrogen mineralization and immobilization. PC affected the microflora in charge of mineralization and enzyme production [57,59]. The results show an increase in N levels starting in the fourth week, which could be the result of OMW in improving N in soils. In this context, Moraetis et al. [4] noticed that OMW with a total N of 1.53 $g \cdot L^{-1}$ applied directly to the soil, during five years of experimentation, provided an additional annual contribution of total N of around 12% compared to the control soil (not amended by OMW) on the top soil layer 0–10 cm. In contrast, Brunetti et al. [60] showed that total N was unchangeable after the OMW application on soil. On the other hand, the availability of N in various forms (mineral or organic) may depend on the mineralization process, which was found to be better in sandy soils than in clay soils [58].

The phosphorus levels in untreated and treated soils with OMW were essential to show the beneficial input of this effluent as an organic fertilizer. Unlike the control soil, P increased after the dose spreading of OMW on treated soils as reported in the studies of Brunetti et al. [60]. Similarly, Kavvadias et al. [41] showed that the level of the available P in the treated soil increased 28 times compared to that in the control soil. Di Serio et al. [53] noted an increase in P available content in soil after the spreading of OMW in layers 10–20 and 20–40 cm. Di Bene et al. [22] showed that using the same effluent had led to an increase in the available P after 5 days and a decrease after 6 months of amendment. After three successive years of application of raw OMW, Chartzoulakis et al. [25] showed that the level of available P in the soil remains unchanged and not improved.

K was considered an important element for soils and plants. In this investigation, it was useful to discuss the K evolution and the simultaneous change in elements such as: Ca, Mg and Na (data shown in Figure 4). The progress of these elements showed that the opposite phenomenon was recorded after 2 weeks of amendment. At first, K and Na variations decreased while calcium and magnesium signaled an increase. From the 4th week, the changes varied from one element to another. As for K (Figure 3e,f), the

treated soils resumed the variation increase, which was stabilized at the end of the IP (12 weeks). Meanwhile, Na concentration stabilized along the IP and registered relatively low values (Figure 4a,b). However, soil 2 composed of bentonite clay had a different response from soil 1. Hence, the OMW levels of these elements (K, Na, Ca and Mg), the composition of the soil and the experimental conditions could be taken into account to explain the progress of these elements over time. The results confirm the work of several researchers [2,22,25,54,61]. These results agree with the recommendations for utilizing OMW as a mineral fertilizer. Therefore, OMW spreading on soil could be a solution to reduce the use of chemical fertilizers.

The Na changes confirmed the increase in Na concentration by enhancing the dose applied. In fact, Magdish et al. [62] observed that the Na soil concentration increased by applying the doses of 50, 100 and 200 $m^3 \cdot ha^{-1}$ compared to the control soil. Another study recorded that the sodium concentration suddenly increased after the application of 80 $m^3 \cdot ha^{-1}$ of raw OMW and decreased or even remained unchanged during the incubation period [2]. Indeed, soil irrigation with water containing high levels of sodium provided exchange reactions between fixed Na ions and the released Ca and Mg ions [63]. In addition, $CaSO_4$ and $CaCO_3$ neutralized the excess of Na ions in the soil solution [64]. In fact, under the effect of a sodium excess, the soil changed its structure and led to the clay dispersion and subsequently the soil aggregate dispersion. The results of the soil's Ca level are in agreement with those of Piotrowska et al. [29,47], who showed a decrease in soil Ca content after 42 days of incubation. Sierra et al. [65] suggested that the exchange of Ca with K and Na could be the cause of the decrease in Ca content. Indeed, the decrease in Ca levels in the treated soils could be due to microbiological activity that occurred during the spreading of OMW, adsorption of this element by free binding sites or precipitation of this element in carbonate [66]. For the Mg progress, Moraetis et al. [4] noticed a drop in Mg contents at the soil layer 0–10 cm treated with OMW in a cornfield compared to that of the control soil. In addition, Piotrowska et al. [47] observed the same effect after 42 days of incubation. An increase in Mg concentration was registered after 3 years of treatment at horizons ranging from 0 to 80 cm of soils receiving the gradual doses of 50, 100 and 200 $m^3 \cdot ha^{-1}$ [62]. The clay type and proportion could be determining the soil chemical exchanges and play an important role in soil chemical stability.

For $Cl^-$ changes in soil, Kavvadias et al. [41] reported that soil $Cl^-$ concentrations were high in a 0–25 cm layer of OMW storage basin and gradually declined as the distance from the basin increased. Furthermore, a soil treatment with municipal wastewater rich in chloride increased the amounts of this component [67]. Therefore, the increment of soil chloride content may be attributed to the presence of this ion in significant quantities in OMW. In addition, Kavvidas et al. [41] noticed that sulfate contents increased at a layer 0–25 cm of OMW storage basin compared to control soil and decreased in deeper horizons. In fact, soil sulfate levels may be reduced as a result of $SO_4^{2-}$ adsorption by clays, colloids and oxides of iron and aluminum [68]. This decrease was the result of sulfate uptake by soil microorganisms or the sulfate was reduced to form hydrogen sulfides by heterotrophic microorganisms.

Rate of the soil organic matter is key for a healthy and high-quality soil. During the incubation, Gargouri et al. [69] mentioned that SOM levels decreased after one month of OMW spreading when compared to control soil levels but then returned to high levels when compared to the reference soil. In fact, this diminution over the incubation period was the result of the soil microorganism activity and that of the OMW. Admittedly, most of the organic fraction of OMW was composed of easily biodegradable entities such as sugars, proteins [70] and other less biodegradable molecules: polysaccharides, polyalcohols and polyphenols [71]. The presence of easily biodegradable compounds in OMW enhanced the microbiological activity and mineralization of SOM as described by Bustamanti et al. [57]. Other studies showed that the clay influenced the mineralization of SOM. That means the clay was able to preserve the organic matter and prevent its mineralization [58]. Several researchers noted an increase in SOM in the treated soil compared to the control soil [28,62,72].

The SOM rates were improved by amending the soil with a high amount of organic carbon. This directly influenced soil characteristics through interdependent biological, chemical and even physical modifications [73]. In the long term, the result of combinations between different soil elements and OMW entities (suspended matter, soluble organic matter and mineral salts) changed the distribution of soil pores and increased the field capacity [72]. The entities and elements previously mentioned played a significant role in linking the various soil aggregates [28]. This contribution enriched the SOM content, resulting in improved water retention, soil stability structure and a high CEC. The decrease in the E2/E6 ratio was linked to the microbiological activity [3]. The E4/E6 ratio was mostly considered as the humification index. The decrease in this ratio was the result of the mineralization of carbohydrates and quinones, as well as the oxidation of phenolic compounds. The latter involved the aliphatic chains of humic substances. During the soil incubation period, the E2/E6 and E4/E6 ratios decreased, indicating an increase in humified organic matter and revealing the humification of organic materials. The ratios of treated soils indicated that high-humic substances could be characterized by aromatic polymers or polycondensate structures ensuring better soil stability.

PC progress in soil at the beginning of the OMW amendment showed an amplification of the level as also observed by Sierra et al. [6]. In addition, the amendment of wine industry wastewater on soil showed an increase in PC concentrations at the start of the application [57]. This improvement could be the result of the OMW spreading which contains a significant amount of PC. Sierra et al. [6] reported a drop in PC levels in treated soils over time to join the control soil values. This group noted that from the 17th day of applying 30 $m^3 \cdot ha^{-1}$ of OMW on soil, the PC concentrations were similar to those of the control soil. This decrease could be the result of the adsorption of PC from OMW by the soil [41,74]. Moreover, Di Bene et al. [22] mentioned that PC underwent soil degradation. Indeed, these researchers showed that PC levels for the treated soil with a dose of 80 $m^3 \cdot ha^{-1}$ of OMW were significantly higher than PC levels in the control soil in the 0–20 cm horizon. After six months, the PC level decreased and reached the level of the untreated soil. Caravaca et al. [74] reported that clays were characterized by a highly specific surface able to adsorb humic substances and hence improve the soil aggregates' stability [75]. Following the first application of OMW on soil, Zenjari and Nejmeddine [76] found that clay with a high adsorption capacity could remove 99% of minerals and phenols, but this capacity decreased after the second application, indicating phenol migration to deep layers. Furthermore, soil clay had a significant impact on organic matter transformations and increased the concentration of some recalcitrant polyphenols [20,77]. In addition, Di Serio et al. [53] indicated that the soil phenol content decreased with the activity of some bacteria and fungi. Therefore, the PC decrease may be due to their adsorption by soil or the result of the condensation and polymerization of humic substances. The degradation of PC generated metabolizable compounds and reduced phytotoxicity. In fact, the metabolism of these polymers required the presence of several enzymes such as Mn peroxidase, hydrolases and phenol hydrolases. The enzymes biosynthesized by bacteria and fungi could have been occurring for depolymerization [78]. Then, decarboxylation and demethylation reactions led to aromatic derivate formation. In addition, the interactions between molecules during kinetics adsorption may depend, in some cases, on the molecule size and diffusion soil solution based on the adsorption that occurred between organic compounds and soil.

Referring to Paredes et al. [79], GI > 50% indicated that the amended soil did not affect the plant; however, other researchers judged that the GI between 66 and 100% was non-phytotoxic [80]. GIA exceeded 150% in this study, and this was an indicator of the soil's high content of fertilizing nutrients. After the first and third annual applications of OMW doses of 50, 100 and 200 $m^3 \cdot ha^{-1}$ on soil, Magdish et al. [26] found that the GI of radish seeds (*Raphanus sativus* L.) was greater than 110% compared to that of untreated soil. Indeed, Piotrowska et al. [47] mentioned that during incubation (42 days), there was an increase in the GI percentage indicating a reduction in phytotoxicity. El Hadrami et al. [81] indicated that OMW reduced the germination of Mediterranean species such as chickpea

(*Cicer arietinum* L.), durum wheat (*Triticum durum*), maize (*Zea mays* L.) and especially tomato (*Lycopersicon esculentum*). The reduction in the GI, according to these researchers, was due to the effect of polyphenols, which differed depending on the species used. The presence of phenols and other organic compounds, as well as the EC and pH of OMW, may cause a drop in GIA and GIT percentages [82]. The same observation was reported for seeds of *Lepidium sativum* L. and *Lactuca sativa* L. according to Pierantozzi et al. [83]. Ouzounidou et al. [84] showed that tomato roots were sensitive to OMW and this was under the effect of fats and polyphenols. Additionally, Buchmann et al. [85] established a clear relationship between OMW phytotoxicity and PC levels. These compounds affected the growth of *Lepidium sativum* but not their germination. For the same species, Greco et al. [86] mentioned that OMW phytotoxicity could be reduced by the oxidative polymerization of monophenolic compounds. Therefore, OMW doses applied on the soil did not affect the GIA and GIT percentages similarly. The OMW could have selective phytotoxicity depending on the species used.

## 5. Conclusions

The use of OMW as a liquid fertilizer improves the soil characteristics by increasing SOM, P and N levels. K concentrations increase throughout the incubation. The parameters previously cited are essential to guarantee an improvement in soil fertility. The increment at IP = 0 weeks of EC, Na and $Cl^-$ for treated soil is temporary, and it was reduced at the end of the experiment. The PC decreases during the incubation. The OMW dose of 40 $m^3 \cdot ha^{-1}$ applied on soils develops a similar progress as control soils. The structure and properties of the clay have a direct effect on the impact of OMW once applied on soil.

**Author Contributions:** Conceptualization, L.C. and M.K.; methodology, L.C.; software, L.C.; validation, T.M. and M.K.; formal analysis, L.C.; investigation, L.C.; data curation, L.C.; writing—original draft preparation, L.C.; writing—review and editing, T.M. and F.B.R.; visualization, L.C.; supervision, T.M. and M.K.; project administration, M.K.; funding acquisition, N.S.A. and A.A. All authors have read and agreed to the published version of the manuscript.

**Funding:** This research was funded by the Deanship of Scientific Research at King Khalid University under grant number RGP. 2/57/43, Princess Nourah bint Abdulrahman University Researchers Supporting Project (number PNURSP2022R19) and Princess Nourah bint Abdulrahman University, Riyadh, Saudi Arabia.

**Institutional Review Board Statement:** Not applicable.

**Informed Consent Statement:** Not applicable.

**Data Availability Statement:** The data presented in this study are available upon request from the corresponding author.

**Acknowledgments:** The authors extend their appreciation to the Deanship of Scientific Research at King Khalid University for funding this work through the Research Groups Program under grant number RGP. 2/57/43, as well as Princess Nourah bint Abdulrahman University Researchers Supporting Project (number PNURSP2022R19) and Princess Nourah bint Abdulrahman University, Riyadh, Saudi Arabia. The authors are particularly very grateful for the financial and administrative support of the Tunisian Ministry of Higher Education and Scientific Research.

**Conflicts of Interest:** The authors declare no conflict of interest.

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
