# Peer review of "Soil Responses to High Olive Mill Wastewater Spreading"

_agronomy, doi:10.3390/agronomy12040972_

Round 1

Reviewer 1 Report

The manuscript entitled “Soil responses to high olive mill wastewater spreading” is well written and easy to follow. The experiment is properly designed,  the results are well described. The discussion is comprehensive. The conclusions are well drawn.

Major remarks:

  1. Introduction is written clearly, however more information about volume of olive mill wastewater depending on the milling method and the levels of fats, organic matter and especially of contamination should be added.
  2. In Methods, please add the information if the analysis of chemical composition of incubated soils were done at day “0” i.e. day when the OMW was added.
  3. In Methods, although the reference for artificial soil was added please add the constituents of the soil, because in results in Table 2 there is soil 1 and 2 and peat.
  4. Why was the artificial soil used in the incubation experiment? Was it this soil similar to typical soils in Tunisia on which the OMW is spread?
  5. How was the soil prepared in terms of moisture? Why moisture was not measured? In most incubation experiments water in soil is maintained at 50-60% of water-holding capacity.
  6. The ANOVA could help in interpretation of the results. The differences between dates or soils or doses would be endorsed by statistical analysis. This analysis would also reveal which factor influences the differences.
  7. In 3.2.1. Effect of OMW on pH soil and EC progress, the description of EC changes is very confusing and hard to follow. I suggest to re-write this part and describe or the doses separately or the soils.
  8. In most incubation experiments with organic fertilizers the volatilization of nitrogen into NH3 is negligible. On the other hand the nitrification should not change total N. The analysis of nitrate nitrogen and ammonium could answer this question.
  9. How could you explain the evolution of total P, especially the decrease in first two weeks and the increase in weeks 4 to 8 with the decrease in last the week 10? The similar question is addressed to the evolution of K.

Minor remarks:

Units according to Instruction for Authors should be expressed as SI units. According to the 9th edition of The International System of Units (SI) given by Bureau International des Poids et Mesures the quotients of SI units should be expressed using either a solidus (/) or a negative exponent (−). For example: mS/cm = mS cm1. Please use this expression instead of for example: mS.cm-1. Please unify this unit because in Table 1 is written as mS.cm-1 but in text is written as μS/cm. Please decide or to use solidus (?) or a negative exponent (-) in text, tables and  graphs.

Line 107 and 109: do these two sentences contain the same information?

Line 156: please add unit for PC value

Line 248: please check the comma in “m (Table 1) ,subse-“

Line 249: there should be a space between “1” and “was”

Line 453: please add a space after point and before  “Several”

Line 501: please add the space after point

Reviewer 2 Report

Manuscript title “Soil responses to high olive mill wastewater spreading” is consist quit good observation. Paper could be very interesting for the plant Biologist if your data could be more consisted and on ecological significance. I find lots of information related to materials and methods also in results consist in the paper not directly focussing on importance of the study.

  1. Abstract line 20-21 need to rewrite, Abstract should have crisp information about the aim of writing this paper
  2. Introduction kindly write all information with facts and value with recent references.
  3. OMW issues increase in winter but sample collected in summer. Kindly, justify.
  4. Line 131-132 why these plant seeds were used in the study, these plant specification not mentioned in the introduction.
  5. Paper findings are good and need to discuss properly.
  6. Study should consist germination and growth of mono and dicot plants (Selectively) and validate your results with advance findings.
  7. Ecological significance of the study is very important and need to be focused.
